# Multi-Modal Imaging to Assess the Follicular Delivery of Zinc Pyrithione

**DOI:** 10.3390/pharmaceutics14051076

**Published:** 2022-05-17

**Authors:** Sean E. Mangion, Lydia Sandiford, Yousuf Mohammed, Michael S. Roberts, Amy M. Holmes

**Affiliations:** 1Therapeutics Research Centre, UniSA—Clinical and Health Sciences, University of South Australia, Adelaide, SA 5000, Australia; sean.mangion@mymail.unisa.edu.au (S.E.M.); lydia.sandiford@kcl.ac.uk (L.S.); m.roberts@uq.edu.au (M.S.R.); 2Basil Hetzel Institute for Translational Health Research, Woodville South, SA 5011, Australia; 3Sydney Medical School, University of Sydney, Camperdown, NSW 2006, Australia; 4Therapeutics Research Group, The University of Queensland Diamantina Institute, Faculty of Medicine, University of Queensland, Brisbane, QLD 4102, Australia; y.mohammed@uq.edu.au

**Keywords:** multiphoton imaging, fluorescence lifetime imaging, follicular delivery, Zinpyr-1, seborrheic dermatitis

## Abstract

Zinc pyrithione (ZnPT) is a widely used antifungal, usually applied as a microparticle suspension to facilitate delivery into the hair follicles. It then dissociates into a soluble monomeric form that is bioactive against yeast and other microorganisms. In this study, we use multiphoton microscopy (MPM) and fluorescence lifetime imaging microscopy (FLIM) to characterise ZnPT formulations and map the delivery of particles into follicles within human skin. To simulate real-world conditions, it was applied using a massage or no-massage technique, while simultaneously assessing the dissolution using Zinpyr-1, a zinc labile fluorescent probe. ZnPT particles can be detected in a range of shampoo formulations using both MPM and FLIM, though FLIM is optimal for detection as it allows spectral and lifetime discrimination leading to increased selectivity and sensitivity. In aqueous suspensions, the ZnPT 7.2 µm particles could be detected up to 500 µm in the follicle. The ZnPT particles in formulations were finer (1.0–3.3 µm), resulting in rapid dissolution on the skin surface and within follicles, evidenced by a reduced particle signal at 24 h but enhanced Zinpyr-1 intensity in the follicular and surface epithelium. This study shows how MPM-FLIM multimodal imaging can be used as a useful tool to assess ZnPT delivery to skin and its subsequent dissolution.

## 1. Introduction

Zinc pyrithione (ZnPT) is an antifungal compound used topically to treat dandruff and seborrheic dermatitis (SD) [1], common scalp conditions estimated to affect up to 50% of the population [2]. *Malesezzia* yeast have been implicated in the development, progression and recurrence of dandruff and SD [3]. The primary mode of action of ZnPT is related to its antimycotic properties against *Malassezia* with a minimum inhibitory concentration of 10–15 ppm [4]. The effective delivery of ZnPT from formulation to the scalp is critical for its efficacy. ZnPT becomes bioactive once it is solubilised in skin lipids (e.g., sebum, intercellular lipids) [5]; therefore, understanding the spatial deposition of particles and dissolution is important for optimising therapy. Furthermore, it has garnered increasing interest over the last several years and most recently, in March 2022, received tighter regulations on its use in the European Union [6].

The target sites for ZnPT include the superficial layers of the stratum corneum [7] and the recessed microenvironments of the scalp hair follicles where yeast reside [8]. Because the zone of antimycotic activity surrounding ZnPT particles is limited by its rate of dissolution and the replenishment of lipids after washing, ZnPT should be deposited evenly over the skin surface [9]. The follicle is an ideal microenvironment for *Malessezia* to flourish [10] because it is lipid rich and shielded from corneocyte sloughing. Follicular delivery is not only important for treating existing dandruff but preventing its recurrence because even in the face of effective surface yeast elimination, the infundibular populations can recolonise the scalp forming the basis for continued symptoms.

Previous methods for detecting follicular ZnPT include confocal microscopy [10], autoradiography [11] and Raman microscopy [12]. Whilst these are important techniques and potentially complementary to the multimodal imaging technique described here, they have some important disadvantages [4]. Reflectance confocal microscopy suffers from the limitation of being non-specific for ZnPT, meaning debris and other ingredients deposited on the scalp may interfere with detection. Autoradiography relies on expensive radiolabels for investigation, while Raman has low spatial resolution and can lack sensitivity in complex biological samples.

We have previously demonstrated that ZnPT particles display luminescence ideal for detection on the skin using fluorescence lifetime imaging microscopy (FLIM) [13]. In complex tissues, such as skin containing endogenous fluorophores (e.g., keratin, collagen, elastin and melanin) [14,15], FLIM has the advantage of providing highly characteristic fluorescence lifetimes [16] enabling ZnPT to be distinguished from endogenous components. Combined with multiphoton microscopy (MPM), this multimodal approach provides high-resolution spatial information about ZnPT delivery. Furthermore, MPM-FLIM can be used for in vivo investigations, especially important for exploring delivery under in-use conditions, like those we have conducted for ZnO nanoparticles in sunscreen assessing the effect of occlusion [17], massage [18], bathing [19] and repeated application [20].

In this work, we characterise the luminescence and microstructure properties of ZnPT in formulations and visualise the follicular delivery under both massage and no-massage conditions. Furthermore, Zinpyr-1 is used to label free zinc within the skin as an indirect measure of ZnPT dissolution, similar to previous work using Zinpyr-1 to detect elevated zinc in skin topically treated with ZnPT [21] and ZnO [20,22]. Together, these data provide critical information for assessing therapeutic efficacy and safety.

## 2. Materials and Methods

### 2.1. Formulations

ZnPT (Sigma Aldrich, Castle Hill, Australia) was used for characterisation and delivery experiments. ZnPT was suspended in Milli-Q water (Bedford, MA, USA) at 1 or 2% *w*/*v*. Three antidandruff shampoos were used, including Head and Shoulders (H&S) ‘Clean and Balanced Shampoo’ ZnPT 1% *w*/*v* (Proctor and Gamble, Cincinnati, OH, USA), Neutrogena T/Gel ‘Daily Control 2 in 1 Shampoo plus Conditioner’ ZnPT 1% *w*/*v* (Johnson and Johnson, New Brunswick, NJ, USA) and Cedel ‘Anti-dandruff Medicated Shampoo’ ZnPT 2% *w*/*v* (Brands RMJ, Victoria, Australia). Full ingredients are available in Appendix A.

### 2.2. Spectral Characterisation

Formulations were imaged using a tuneable titanium sapphire Mai Tai laser (80 MHz, <200 fs, Spectra Physics, Mountain View) capable of two-photon excitation 710–910 nm. A spectral emission scan was performed on a Zeiss LSM 710 microscope (Carl Zeiss Microscopy GmbH, Berlin, Germany). Zen Black 2010 version 1.1.2.0 (Carl Zeiss Microscopy GmbH, Berlin, Germany) was used to extract intensity data, before manual correction for caprylic capric triglyceride (CCT) (Dow Chemical Company, Frenchs Forest, Australia) as background control and normalisation for laser power.

### 2.3. Cryo-Scanning Electron Microscopy (Cryo-SEM)

Cryo-SEM was performed on a JEOL (JSM 7100F) (JEOL Ltd., Tokyo, Japan). Samples were plunge-frozen under vacuum using a slush nitrogen (−210 °C), before being moved quickly to the cryo-preparation chamber, fractured and sputter-coated with platinum (120–240 s, 10 mA). The coated samples were moved to the imaging chamber maintained at −145 °C (anti-contaminator −194 °C). Images were collected using the secondary electron and backscatter signal (7 kV, working distance 10 mm). For energy dispersive X-ray spectroscopy (EDS), a 1 nA probe current was used for a survey spectrum 0–20 keV. These images were acquired at a voltage of 20 kV for elemental analysis.

### 2.4. Human Skin Experiments

Full-thickness human skin was donated by a 46-year-old female patient undergoing abdominoplasty. Ethics approval was obtained from the Human Research Ethics Committee of the Queen Elizabeth Hospital in Adelaide (reference number 2009208) and the patient provided full written consent.

Aqueous ZnPT 2% *w*/*v* and commercial shampoos were applied (10 mg/cm^2^, total dose area 1.3 cm^2^) either with no massage or with a 2 min massage by an electric toothbrush (head was wrapped in parafilm to prevent the loss of formulation in the bristles). Skin samples were mounted to Franz diffusion cells with PBS receptor and incubated in a water bath at 35 ± 1 °C for 24 h. Skin was then embedded in optimal cutting temperature (OCT) media (Grale HDS, Ringwood) and snap frozen at −80 °C. The OCT-embedded skin was processed into 20 µm transverse cross-sections using a cryostat at −20 °C (Leica CM 1950, Victoria, Australia).

### 2.5. Multiphoton Microscopy and Fluorescence Lifetime Imaging (MPM-FLIM)

Full-thickness skin set aside for *en face* imaging was stained with acriflavine hydrochloride (Sigma-Aldrich, Castle Hill, Australia) 100 µM in water overnight. Skin cryosections were stained with 10 µM Zinpyr-1 (Cayman Chemical Co., Ann Arbor, MI, USA).

For multiphoton microscopy (MPM), the 488 nm argon laser was used to excite Zinpyr-1, while the 800 nm tuneable titanium-sapphire Mai-Tai was used to excite ZnPT. Emission was collected from the descanned line under a filter (520–560 nm) for single-photon excitation and a non-descanned filter (395–420 nm) for two-photon excitation. A 40X water immersion objective was used.

Fluorescence lifetime imaging microscopy (FLIM) took place along the non-descanned line of the Zeiss LSM710 microscope, fitted with two bh GaAsP hybrid detectors HPM100-40 and two TCSPC modules SPC-152 (Becker & Hickl GmbH, Berlin, Germany). The samples were excited at 740 and 800 nm. Emission was collected using a 405/10 and 540/20 nm bandpass filter (Semrock Inc., Rochester, NY, USA). Images were captured over 5 min. For *en face* imaging, acriflavine-stained skin was imaged from surface, 40 and 90 µm using the same FLIM parameters as for Zinpyr-1.

Zin-pyr-1 cryosections were first imaged by MPM using the tile-scan function to capture an entire follicle. FLIM images were then acquired along the length of the follicle for detection of ZnPT (λ_exc_ = 740 nm).

### 2.6. Dissolution Studies

ZnPT (0.4 mg/mL) was placed into an Ibidi well chamber slide with either olive oil (home brand extra-virgin), artificial sweat (Pickering labs, Mountain View, CA, USA), DMSO (Sigma Aldrich, Castle Hill, Australia) or water. The wells were imaged at 0 h and then incubated for 18 h at 32 °C on a shaker bath. MPM and FLIM images were then acquired at 18 h.

### 2.7. Data and Image Analysis

After acquisition, FLIM images were analysed using SPCimage software version 5.2 (Becker & Hickl GmbH, Berlin, Germany). For assessment of the FLIM characteristics of the formulations, 5 pixels were chosen at random from each image and the average ± SD was calculated for: *τ*_m_, *τ*_1_, *τ*_2_, *α*_1_, *α*_2_. For analysis of particle size, ImageJ version 1.50 (National Institutes of Health, Bethesda, MD, USA) was used. MPM images of the formulations were first thresholded prior to particle analysis using the inbuilt particle analyser function for Ferets diameter. For area analysis, ImageJ function was used in the same manner. Both MPM and FLIM images were used for evaluating ZnPT delivery depth in the follicles. Firstly, if ZnPT was present, it was identified by its characteristic lifetime. Then, the corresponding location was identified in the MPM image, and the depth was measured from the skin surface to this point along the follicle using ImageJ. The deepest deposited particle was used to define the delivery depth. Delivery depths of ZnPT were determined in 3 different follicles for each condition. Where appropriate, results are presented as mean ± SD. GraphPad Prism (version 8.4.3, San Diego, CA, USA) was used to graph results and perform Student’s *t*-test for follicular delivery depth.

## 3. Results

### 3.1. Spectral Characterisation of Commercial Formulations

The transmission images show commercial products contain well-dispersed micron-sized particles (Figure 1A). The merged images demonstrate that luminescence under two-photon excitation (800 nm) originates from these particles. There is an overlap in emission for the different formulations, with a peak emission at 400 nm corresponding to ZnPT (Figure 1B). Meanwhile, the peak at 650 nm can be attributed to excipients.

### 3.2. Time-Resolved Imaging

The particles can be observed using FLIM (Figure 1C) with a similar morphology as fluorescence imaging. Particles had a short time-weighted lifetime, as demonstrated in the corresponding decay curve (Figure 1D). The average values and component lifetimes are shown in Figure 1E. The lifetime values are similar across all formulations, ranging from 172 to 193 ps. These short lifetimes were used to distinguish ZnPT particles from endogenous skin components (e.g., collagen, keratin and lipofuscin) in the proceeding work.

### 3.3. Microstructure and Particle Size

Cryo-SEM was performed on the shampoos to visualise the microstructure in the hydrated state (Figure 2A). In H&S, two different particles were visible, including multiple smaller particles 2–4 µm in size (Particle A) and a single 12 µm particle (Particle B). An elemental analysis (Figure 2B) revealed that Particle A had a high abundance of zinc, magnesium and carbon. Particle B had a high abundance of silicone, oxygen and carbon. In the cryo-SEM image of the T/Gel, no particles were visible. Many particles were visible in the cryo-SEM image of the Cedel. These particles could be observed either in clusters of approximately 5 µm or singularly with a particle size of 1–2 µm displaying a hexagonal plate shape. There also appeared to be finer particles in the submicron size range that were closely located to these plates.

Further analysis of the images collected under the two-photon excitation proved useful for particle size analysis, with the advantage of a significantly increased sample size compared to cryo-SEM (Figure 2C). The average particle sizes were: 7.2 ± 7.7 µm (ZnPT aq), 3.3 ± 1.8 µm (Cedel), 1.6 ± 0.9 µm (H&S) and 1.0 ± 0.4 µm (T/Gel) (Figure 2D).

### 3.4. Dissolution of ZnPT

The dissolution of solid-state ZnPT is required for bioactivity. The effect of a solvent on dissolution was studied by MPM and FLIM imaging using water, olive oil (to simulate the high triglyceride and fatty acid content in sebum [23]) and artificial sweat. Particle dissolution was most apparent for ZnPT in oil and sweat, detected using MPM (*p* < 0.001, Appendix A) over 18 h at 32 °C. This trend was mirrored in the more sensitive FLIM imaging method. The DMSO served as a positive control, with no particles visible. No significant changes in the particle amount were observed for water. FLIM imaging also demonstrated no significant changes in lifetime parameters for the different solvents over the 18 h incubation.

### 3.5. Detection of ZnPT in Hair Follicles from ZnPT (aq)

Two techniques were used for assessing ZnPT delivery into the hair follicles (see Supplementary Material for further description on optimisation). In brief, the less discriminatory, but faster MPM technique was used to acquire mosaics for greater image sizes. This was important for imaging the entire length of the cryosectioned follicle for analysing the ZnPT delivery. The FLIM technique enabled a more sensitive and specific analysis along the follicle for a time-resolved detection of the ZnPT.

The delivery of the ZnPT for the positive and negative controls is shown in Figure 3. The MPM images obtained used four individual stitched images to show the length of the follicle obtained via cryosectioning. For the blank negative control follicle, no ZnPT was detected in the corresponding FLIM images after the lifetime analysis. For the positive controls, aqueous ZnPT 2% *w*/*v* was topically applied to the skin. In the no-massage follicle, ZnPT can be detected in the corresponding FLIM images at a depth of 250 µm. In the massage follicle, ZnPT particles can be detected up to a depth of 500 µm. While there was an increase in the delivery depth for massage, differences were not significant (Figure 3G, *p* = 0.63).

### 3.6. Detection of ZnPT in Hair Follicles from Commercial Products

The delivery of the ZnPT from the H&S shampoo after no massage and a 2 min massage is shown in Figure 4. The two MPM images are composed of 25 stitched images to show the length of the follicles obtained via cryosectioning. For the no-massage application, the ZnPT was visible in the upper infundibulum to a depth of 90 µm. In the remaining three follicles for the no-massage application, the ZnPT could not be detected (see Appendix A for representative images). For the massage application, the ZnPT was deposited at 75 µm in the infundibulum. In the remaining two follicles acquired, no ZnPT was detected. A notable feature of the no-massage application was what appeared to be a keratin plug occluding the entrance of the follicle.

Table 1 summarises the findings for all the follicles investigated. For each condition, three follicles were chosen to study the delivery depth. For the commercial formulations, less consistent results were observed, that is, in most cases, none or minimal particles were observed surrounding the follicle entrance. In all the treatments, particle delivery to the follicles was not evenly distributed (i.e., they were deposited in clumps, not evenly spanning the follicle).

### 3.7. ZnPT Dissolution on Skin Surface and Within Hair Follicles

The Zinpyr-1 was used to map the labile zinc ion levels in the skin following the ZnPT application. The intensity levels of this probe were determined across the profiles of the skin on both the skin surface (interfollicular) and within the follicles for the ZnPT applied with massage (Figure 5A,B). In this way, it was possible to indirectly assess where the ZnPT had dissolved. The Zinpyr-1 intensity levels were similar in the follicular skin treated with ZnPT (aq) and the control; however, they were elevated in the viable epidermis of the ZnPT-treated skin (Figure 5C). The greatest intensity was observed for the viable epidermis from the interfollicular skin treated with ZnPT (aq). Meanwhile, for the commercial formulations, the interfollicular skin also had the greatest Zinpyr-1 signal compared to the follicles (Figure 5D). The T/Gel showed the greatest signal compared to the Cedel and H&S (Figure 5E) in both the follicle and interfollicular skin.

## 4. Discussion

Our group previously reported that ZnPT possesses luminescence and that it can be identified on the skin using its characteristic lifetime [13]. This work is an extension of these investigations and aimed to explore the follicular delivery of ZnPT from several commercial antidandruff shampoos. It should be noted that the purpose was not to compare the effectiveness of the products, but rather to assess MPM-FLIM as a technique for measuring skin delivery.

The spectral characterisation of the formulations (H&S, Cedel and T/Gel) confirmed the presence of ZnPT with emission maxima at 380–400 nm consistent with our previous report [13]. An additional broad emission peak at 600–700 nm can be attributed to excipients within the formulation. The lifetime measurements obtained via FLIM also confirmed the presence of ZnPT particles with a short lifetime in the range of 200 ps. In commercial formulations, this was reduced to approximately 180 ps, likely due to the different microenvironments of these formulations. FLIM is solvent-dependent and highly sensitive to factors in the local environment, including viscosity and pH [24], and the degree of neighbour-particle crowding [25].

Particle morphology is a factor known to influence topical delivery [26,27,28,29]. Two-photon fluorescence provided images to assess particle size. Direct light-scattering methods for assessing size distribution were not appropriate in this case because they rely on a pure and non-sedimenting sample with the assumption that all the particles are perfect spheres [30]. The ZnPT particles were platelet in shape as revealed by cryo-SEM. The shampoos differed to the aqueous ZnPT as the particles were finer (Cedel > H&S > T/Gel) and existed as a stabile dispersion [31,32,33]. Industry reports suggest that the optimal ZnPT particle size for surface delivery is approximately 2.5 µm which is similar to the range we observed of 1.0–3.3 µm [34].

Previous methods for assessing the delivery of ZnPT to the skin include autoradiography [11], reflectance confocal microscopy [10,35] and Raman spectroscopy [12]. FLIM improves both the specificity and sensitivity method for identifying ZnPT against the endogenous components of skin (e.g., keratin, collagen and elastin) [16]. Furthermore, tissue sectioning in this work allowed a deep assessment of follicular delivery, which is not achievable with optical-sectioning form reflectance confocal microscopy. The main limitations of the MPM-FLIM method used here is that image acquisition and sample processing can be time consuming and labour intensive (i.e., not ideal as a high-throughput test) and that it relies upon pico-second lasers which are costly.

The greatest extent of particle deposition was observed for the application of aqueous ZnPT, up to approximately 500 µm with the massage application. Fewer particles were observed for the commercial formulations after incubation. They deposited more superficially at approximately 50 µm, consistent with previous imaging studies [10]. These finer, more dispersed particles would have dissolved at a faster rate, supported by our dissolution studies, demonstrating the greatest dissolution in the artificial skin components of olive oil [23] and sweat.

The sensitivity of the FLIM method depends on the detectors used. Increased efficiency has been reported for the GaAsp detectors in this work [36] and is confirmed by the visualisation of micron- and submicron-sized ZnPT particles on the skin surface. These amounts should be quantitated in further work using X-ray fluorescence [21].

Massage did not have a significant effect on the depth of the follicular delivery in this study (*p* = 0.63). Previous work has shown that follicular delivery is dependent on particle size with an optimal size of 600–700 nm [26]. This is smaller than the particle size measured for all the formulations and may explain why massage did not have a significant effect on delivery.

Furthermore, formulation viscosity is an important aspect to the lateral transport kinetics of ZnPT on the skin [37]. We speculate that viscosity was a key factor that may have impacted the flow of ZnPT into the follicles. ‘Active’ follicles are receptive to particles and ‘inactive’ follicles are not receptive to particles [38,39]. Several MPM images appeared to show keratin build-up at the follicle entrance. It is possible that the ZnPT did not penetrate the follicle because this build-up occluded the entrance, rendering it ‘inactive’.

Zinpyr-1 was used as a marker for labile zinc following ZnPT application. This has been shown in previous work for ZnO [22] and ZnPT on heat-separated epidermis [21]. It enabled an investigation of the effects of ZnPT dissolution on the skin, both at the surface and within the follicles. Elevated levels were observed on the surface for skin treated with ZnPT, consistent with imaging data which showed the delivery of particles. The intensity levels of Zinpyr-1 were not elevated compared to the control in the follicle, suggesting that minimal particle deposition and dissolution occurred in this region, supported by the particle deposition mostly at the superficial entrance. For the commercial formulations, the highest intensity was observed for the T/Gel, explained by the smaller particle size which would have resulted in the fastest rate of dissolution, consistent with results from the particle deposition. It should be noted, however, that other sources of zinc, such as the zinc carbonate present in the formulations, may have added to the zinc levels observed.

A limitation of the current study was that the only skin source for the experiments was ex vivo skin from the abdomen. Ideally, scalp skin would have been used; however, it is difficult to source from living donors. The terminal follicles obtained in this study had similar morphometry to those previously reported for follicles on the scalp [40], but the abdominal skin had a significantly lower hair density [38]. Scalp skin from cadavers could be used in future work. If there was a high density, this would reduce tissue processing times and enable more follicles to be assessed.

Pig skin has been widely used for studying follicular delivery [26,41,42]. It has been proposed that pig skin from the ear may be a more appropriate model than excised human skin [41]. This is based on estimates that the follicle reservoir of ex vivo human skin shrinks after excision to about only 10% of the original reservoir due to the contraction of elastin [39]. In pig ear skin, the underlying cartilage supports the follicle and prevents closure after excision [41]. Considering this, our findings could represent an underestimate of the in vivo follicular delivery of ZnPT. While the follicles in pig skin tend to be larger than in humans [40] and often occur in triplicate, the real in vivo delivery could lie somewhere between excised pig ear and excised human skin. Future work should also focus on whether the ZnPT follicular delivery can be optimised by changing aspects of the formulation. ZnPT particles can be synthesised into other shapes, including rods and needles [43]. Mechanochemical milling [44] or sonic methods [45] would most likely need to be used to synthesise particles in the submicron range. Apart from the possibility of enhancing delivery, particles in the nano-size range could result in greater antifungal activity due to the greater surface area for yeast and ZnPT interactions. Formulation factors such as vehicle viscosity [46] are also important factors that warrant further investigation. The effects of repeated application could be studied ex vivo to simulate several weeks of usage with shampoo. Lastly, the MPM-FLIM method presented here can be performed in vivo and investigations should therefore explore follicular delivery to the uppermost follicle after realistic exposure and in-use conditions.

## 5. Conclusions

Understanding the delivery of ZnPT from commercial products is important for assessing both efficacy and safety. This study investigated the luminescence properties and follicular targeting of ZnPT from commercial products, showing that delivery depended on formulation. Particle size was important in determining the extent of the delivery and subsequent dissolution. Massage application did not have a significant effect on the follicular delivery depth. MPM-FLIM is a useful tool that can be used in future work to optimise formulations for improved follicular delivery and sustained dissolution to achieve antifungal effects in dandruff and SD treatment.

## Figures and Tables

**Figure 1 pharmaceutics-14-01076-f001:**
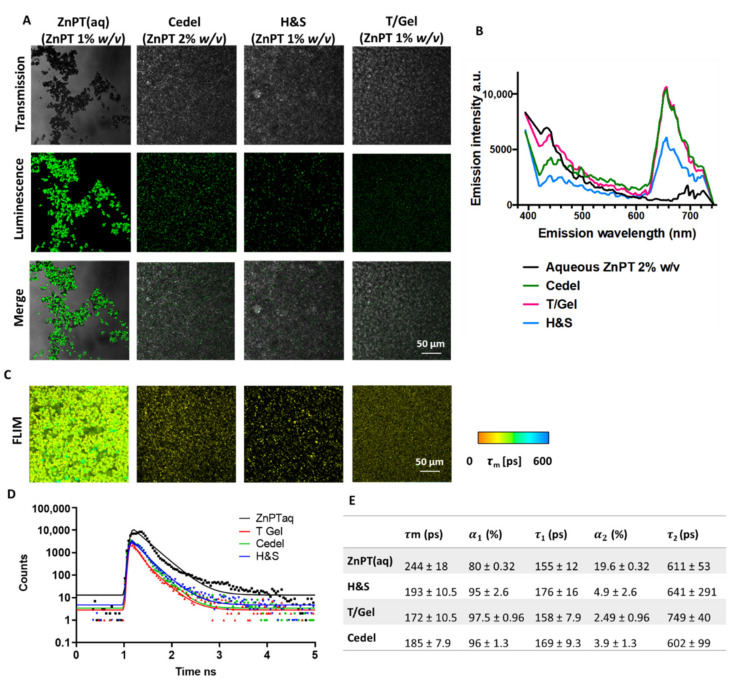
**Spectral and lifetime characterisation of ZnPT particles in formulation**. (**A**) Transmission, luminescence (at 800 nm excitation, emission 395–420 nm) and merge-mode images obtained using 40× objective. (**B**) Spectral emission scan at 800 nm excitation of formulations. (**C**) FLIM images (405/20 nm emission) with (**D**) corresponding decay curves and (**E**) lifetime parameters. τm, average time-weighted lifetime; $\alpha_{1}$, relative-amplitude component of fast-decay component; τ_1_, fast-decay lifetime; $\alpha_{2}$, relative-amplitude component of slow-decay component; τ_2_, slow-decay lifetime.

**Figure 2 pharmaceutics-14-01076-f002:**
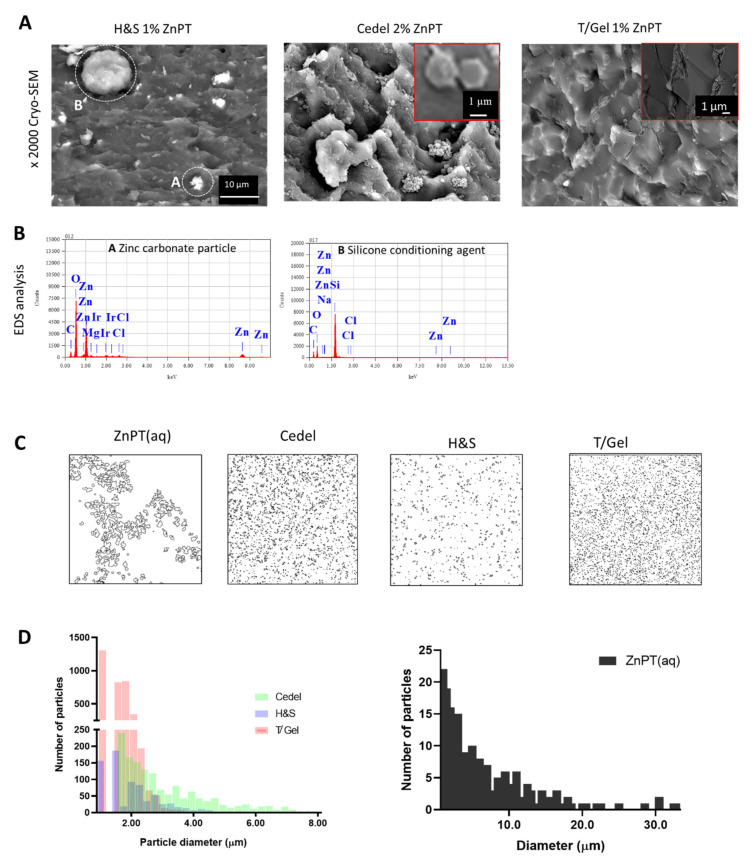
**Microstructure and particle size analysis of commercial formulations**. (**A**) Cryo-SEM images with (**B**) accompanying EDS for Head and Shoulders. Note the different morphologies of the ZnPT particles. In H&S, particles have a platelet shape; for Cedel, more particle agglomeration is evident. T/Gel uses fine particle size ZnPT that is difficult to observe, even at ×10,000. (**C**) Particle outline images used for size analysis, presented in (**D**) showing Ferets diameter for various formulations.

**Figure 3 pharmaceutics-14-01076-f003:**
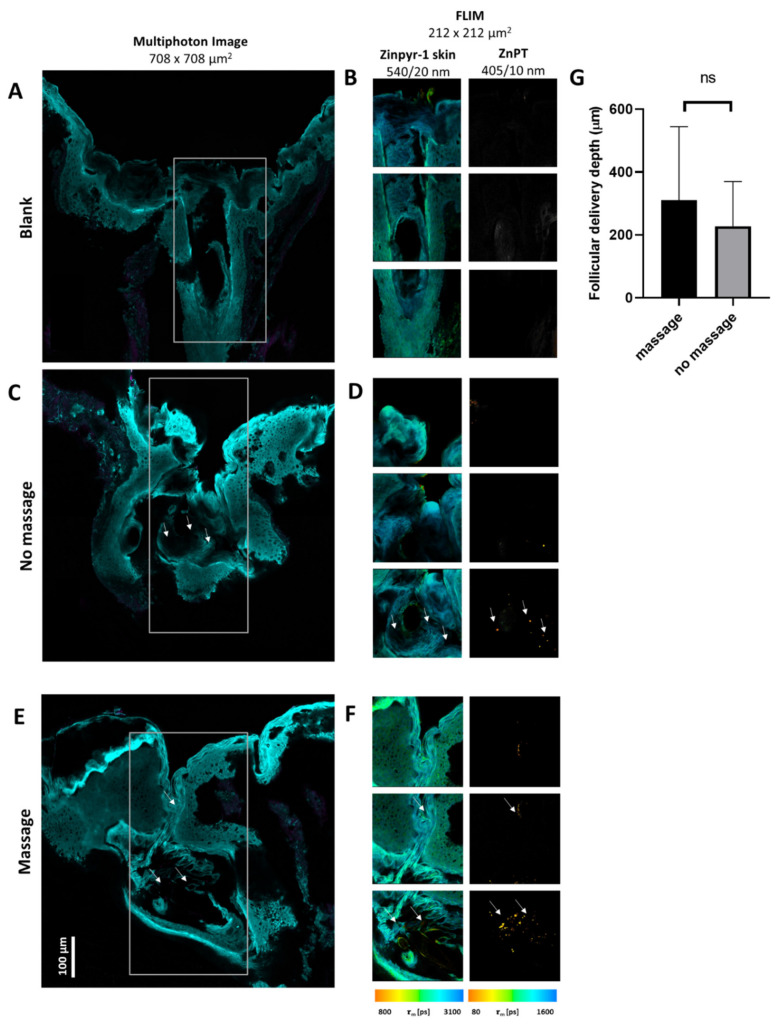
**Follicular delivery of ZnPT from the positive and negative controls**. (**A**) MPM image of a blank negative control with (**B**) the corresponding FLIM images along the length of the follicle. (**C**) MPM image of the no-massage positive control dosed with aqueous ZnPT 2% *w*/*v* and (**D**) the corresponding FLIM images along the length of the follicle. (**E**) MPM image of the 2 min massage positive control dosed with aqueous ZnPT 2% *w*/*v* and (**F**) the corresponding FLIM images along the follicle. For MPM images (**A**,**C**,**E**), two excitation wavelengths were used: two-photon excitation at 800 nm with emission collected at 395–420 nm (pink) and one-photon excitation at 488 nm with emission collected at 520–560 nm (blue). Scale bar = 100 μm. The parameters for the FLIM images (**B**,**D**,**F**): two-photon excitation at 740 nm with emission collected using a 405/10 nm bandpass filter for ZnPT and 540/20 nm bandpass filter for Zinpyr-1. FLIM images are pseudo coloured to the average time-weighted lifetime $\tau_{m}$ [ps]. (**G**) Student’s *t*-test for follicular delivery depth for massage and no-massage application. Arrows indicate position of deposited ZnPT particles.

**Figure 4 pharmaceutics-14-01076-f004:**
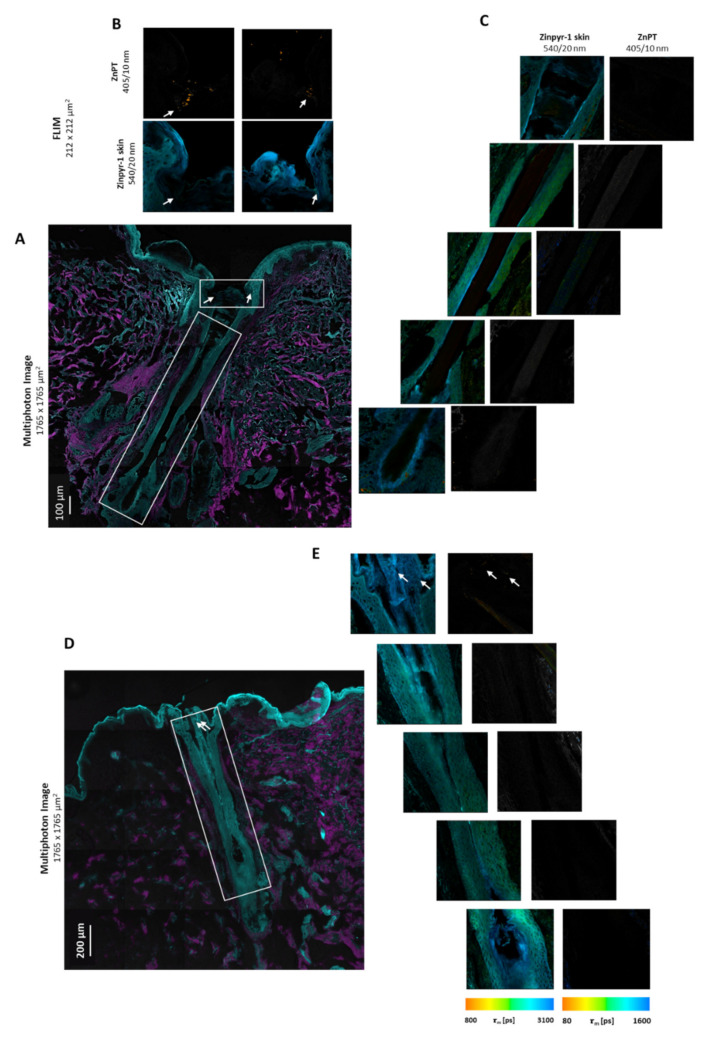
**Follicular delivery from commercial product H&S**: (**A**) MPM image of H&S (ZnPT 2% *w*/*v*) applied with no massage and (**B**) the corresponding FLIM images along the length of the follicle. (**C**) MPM image of DHS applied with 2 min massage and (**D**) the corresponding FLIM images along the length of the follicle. For MPM images (**A**,**C**), two excitation wavelengths were used: two-photon excitation at 800 nm with emission collected at 395–420 nm (pink) and one-photon excitation at 488 nm with emission collected at 520–560 nm (blue). Scale bar = 200 μm (**A**) and 100 μm (**C**). The parameters for the FLIM images (**B**,**D**,(**E**)) were: two-photon excitation at 740 nm with emission collected using a 405/10 nm bandpass filter for ZnPT and 540/20 nm bandpass filter for Zinpyr-1. FLIM images are pseudo-coloured to the average time-weighted lifetime $\tau_{m}$ [ps]. Arrows indicate position of deposited ZnPT particles.

**Figure 5 pharmaceutics-14-01076-f005:**
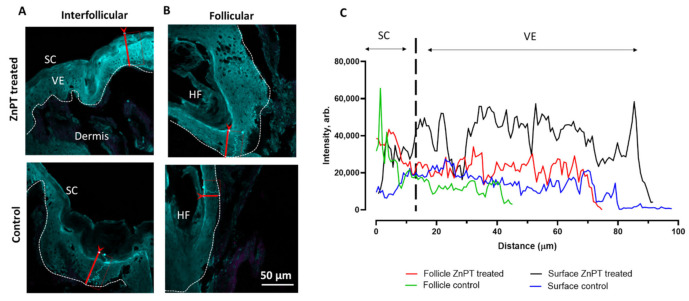
**Elevated labile zinc within the viable epidermis from ZnPT treatment**. (**A**) MPM of inter-follicular/surface skin from control and ZnPT treatment. (**B**) MPM images of follicular skin. (**C**) Corresponding Zinpyr-1 intensity profiles. SC, stratum corneum; HF, hair follicle; VE, viable epidermis. (**D**) MPM of interfollicular and follicular skin from commercial product treatment. (**E**) corresponding intensity profiles (top graph, interfollicular, bottom graph, follicular).

**Table 1 pharmaceutics-14-01076-t001:** Summary of findings from follicular delivery experiments.

Product	Particle Delivery Observations	
	2 min Massage	No Massage
Head and Shoulders (1% *w*/*v*)	Average deposition depth of 57 ± 65 µm (n = 3)	Deposition at 117 µm (n = 1) or not observed (n = 2)
Cedel (1% *w*/*v*)	Average depth 441 ± 543 µm (n = 2) or not observed (n = 1)	Depth of 61 µm (n = 1) or no particles observed (n = 2)
T/Gel (2% *w*/*v*)	Surface deposition (n = 2) or no deposition observed (n = 1)	Deposition at 154 µm (n = 1) or no deposition observed (n = 2)
Aqueous suspension (2% *w*/*v*)	Average deposition depth of 310 ± 243 µm (n = 3)	Average deposition depth of 228 ± 142 µm (n = 3)

## Data Availability

Not applicable.

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
