# Peer review of "Multi-Modal Imaging to Assess the Follicular Delivery of Zinc Pyrithione"

_pharmaceutics, 2022, doi:10.3390/pharmaceutics14051076_

Round 1
Reviewer 1 Report
The manuscript entitled 'Follicular delivery of zinc pyrithione' is not a new topic, but I would like to appreciate the authors for using new techniques like MPM-FLIM.
I have few comments below
- What is the concept behind choosing 34C for dissolution study. In section 3.4. the authors mentioned 32 C.
- Figures 1B,D, are not clear
- Since there are many techniques to measure follicular delivery, please mention them briefly in the introduction part and and explain how MPN-FLIM is superior than those techniques
Author Response
Authors response to reviewers:
Firstly, we would like to thank the comments of reviewers. Their feedback has strengthened our manuscript. Please see below our responses.
Reviewer 1
The manuscript entitled 'Follicular delivery of zinc pyrithione' is not a new topic, but I would like to appreciate the authors for using new techniques like MPM-FLIM.
I have few comments below
- What is the concept behind choosing 34C for dissolution study. In section 3.4. the authors mentioned 32 C.
For the dissolution study 32°C was selected to mimic skin surface temperature. Section 3.4 correctly reflects this, however, as pointed out by the reviewer in Section 2.6 (page 8) there was a typo error of 34°C that has now been corrected.
- Figures 1B,D, are not clear
The text size for these panels has been increased for readability and clarity.
- Since there are many techniques to measure follicular delivery, please mention them briefly in the introduction part and and explain how MPN-FLIM is superior than those techniques.
The following has been added to paragraph 1 on page 5:
“Reflectance confocal microscopy suffers from the limitation of being non-specific for ZnPT, meaning debris and other ingredients deposited on the scalp may interfere with detection. Autoradiography relies on expensive radiolabels for investigation, while Raman has low spatial resolution and can lack sensitivity in complex biological samples.”
This complements what was already in addition to what is already in the introduction:
We have previously demonstrated that ZnPT particles display luminescence ideal for detection on the skin using fluorescence lifetime imaging microscopy (FLIM) [1]. In complex tissues, such as skin containing endogenous fluorophores (e.g. keratin, collagen, elastin and melanin) [2, 3], FLIM has the advantage of providing highly characteristic fluorescence lifetimes [4] enabling ZnPT to be distinguished from endogenous components. Combined with multiphoton microscopy (MPM), this multi-modal approach provides high resolution spatial information about ZnPT delivery.
References-
[1] L. Sandiford, A.M. Holmes, S.E. Mangion, Y.H. Mohammed, A.V. Zvyagin, M.S. Roberts, Optical Characterisation of Zinc Pyrithione, Photochemistry and photobiology, (2019).
[2] H.G. Breunig, H. Studier, K. Konig, Multiphoton excitation characteristics of cellular fluorophores of human skin in vivo, Opt Express, 18 (2010) 7857-7871.
[3] K. Konig, I. Riemann, High-resolution multiphoton tomography of human skin with subcellular spatial resolution and picosecond time resolution, Journal of biomedical optics, 8 (2003) 432-439.
[4] W. Becker, Fluorescence lifetime imaging – techniques and applications, J. Microsc., 247 (2012) 119-136.
Reviewer 2 Report
The manuscript entitled “Follicular delivery of zinc pyrithione”, by Mangion et al have studied MPM-FLIM multi-modal imaging as a useful tool to assess ZnPT delivery to skin and its subsequent dissolution. The work is well presented, and the study design supports the aim and rationale. There are a few suggestions to the authors which would further improve the
readability and usefulness of this work to the readers.
-The title of the manuscript should be improved as it does not convey anything regarding the imaging techniques.
-Limitations of the combined imaging techniques should be mentioned for better clarity to the readers.
-Sensitivity of the imaging techniques should be also discussed as they are limited by staining and fluorophores.
-High resolutions images should be added to the manuscript
Author Response
Authors response to reviewers:
Firstly, we would like to thank the comments of reviewers. Their feedback has strengthened our manuscript. Please see below our responses.
Reviewer 2
The manuscript entitled “Follicular delivery of zinc pyrithione”, by Mangion et al have studied MPM-FLIM multi-modal imaging as a useful tool to assess ZnPT delivery to skin and its subsequent dissolution. The work is well presented, and the study design supports the aim and rationale. There are a few suggestions to the authors which would further improve the
readability and usefulness of this work to the readers.
-The title of the manuscript should be improved as it does not convey anything regarding the imaging techniques.
Agreed, we have now changed the title to: “Multi-modal imaging to assess the follicular delivery of zinc pyrithione”
-Limitations of the combined imaging techniques should be mentioned for better clarity to the readers.
On page 14 in paragraph 1 we have now explicitly outlined the limitations of the technique by adding – “The main limitations of MPM-FLIM method used here is that image acquisition can be time consuming and labour intensive (i.e., not ideal as a high-throughput test), and that it relies upon pico-second lasers which are costly.”
-Sensitivity of the imaging techniques should be also discussed as they are limited by staining and fluorophores.
We have added the following to paragraph 1 on page 14- “The sensitivity of the FLIM method depends on the detectors used for single photon counting. Increased efficiency has been reported for the GaAsp detectors used here [1] and is confirmed by visualisation of micron and submicron sized ZnPT particles on the skin surface. These amounts should be quantitated in further work using x-ray fluorescence [2].”
-High resolutions images should be added to the manuscript
The images provided are within the resolution format requirements set by the publisher, we will check that the file format for review did not impair resolution. In regards to super resolution microscopy we have used structured illumination microscopy [3] in assessing topical delivery of other zinc species such as nano zinc oxide, however it has no advantage in terms of distinguishing exogenous from endogenous components.
References
[1] Increased efficiency by GaAsP hybrid detectors, Microscopy research and technique, 74 (2011) 804-811.
[2] A.M. Holmes, I. Kempson, T. Turnbull, D. Paterson, M.S. Roberts, Imaging the penetration and distribution of zinc and zinc species after topical application of zinc pyrithione to human skin, Toxicology and applied pharmacology, 343 (2018) 40-47.
[3] Z. Khabir, A.M. Holmes, Y.-J. Lai, L. Liang, A. Deva, M.A. Polikarpov, M.S. Roberts, A.V. Zvyagin, Human Epidermal Zinc Concentrations after Topical Application of ZnO Nanoparticles in Sunscreens, Int J Mol Sci, 22 (2021) 12372.
Reviewer 3 Report
ready for publication
Author Response
Thank you to reviewer 3 for taking the time to review our manuscript.